# *Listeria monocytogenes* Modulates Macrophage Inflammatory Responses to Facilitate Its Intracellular Survival by Manipulating Macrophage-Derived Exosomal ncRNAs

**DOI:** 10.3390/microorganisms13020410

**Published:** 2025-02-13

**Authors:** Jian Jiao, Zhongmei Ma, Nengxiu Li, Fushuang Duan, Xuepeng Cai, Yufei Zuo, Jie Li, Qingling Meng, Jun Qiao

**Affiliations:** 1College of Animal Science & Technology, Shihezi University, Shihezi 832003, China; sanningzhi123@126.com (J.J.); shadowxiaoma@163.com (Z.M.); 18899597571@139.com (N.L.);; 2State Key Laboratory of Veterinary Etiological Biology, Lanzhou Veterinary Research Institute, Chinese Academy of Agricultural Sciences, Lanzhou 730046, China

**Keywords:** *Listeria monocytogenes*, exosome, macrophage, inflammatory response, intracellular survival

## Abstract

Exosomes are nanoscale vesicles secreted by cells that play vital regulatory roles in intercellular communication and immune responses. *Listeria monocytogenes* (*L. Monocytogenes*, *LM*) is a notable Gram-positive intracellular parasitic bacterium that infects humans and diverse animal species. However, the specific biological function of exosomes secreted by macrophages during *L. Monocytogenes* infection (hereafter EXO-LM) remains elusive. Here, we discovered that EXO-LM stimulated the secretion of inflammation-associated cytokines by macrophages, facilitating the intracellular survival of *L. monocytogenes* within macrophages. Transcriptomic analysis shows that EXO-LM significantly upregulates immune recognition and inflammation-related signaling pathways in macrophages. Furthermore, a ceRNA regulatory network comprising exosomal ncRNAs and macrophage RNAs was constructed through EXO-LM transcriptome sequencing. Utilizing bioinformatics and dual-luciferase reporter assays, we identified two potential binding sites between lncRNA Rpl13a-213 and miR-132-3p. Cell transfection experiments demonstrated that Rpl13a-213 overexpression augmented pro-inflammatory cytokine expression in macrophages, in contrast to the suppression by miR-132-3p overexpression. The decrease in Rpl13a-213 upon EXO-LM stimulation enhances miR-132-3p expression, dampening the inflammatory response in macrophages and aiding *L. monocytogenes* intracellular survival. This study unveils the immunomodulatory function of exosomal ncRNAs originating from macrophages, which provides fresh perspectives into the mechanisms underlying macrophage inflammatory response regulation by *L. monocytogenes*-infected cell-derived exosomes.

## 1. Introduction

Exosomes are extracellular vesicles, ranging in diameter from approximately 30 to 150 nanometers, secreted by living cells via the multivesicular body (MVB) pathway [1,2]. These phospholipid bilayer-enclosed vesicles harbor a complex cargo of proteins, lipids, nucleic acids, and other molecules, serving as mediators of intercellular communication [3]. As such, exosomes are recognized as one of the most critical mediators of intercellular communication [4,5]. Existing studies have revealed that, during bacterial infections of host cells, the composition of exosomes secreted by these cells undergoes significant alterations [6]. Among the components carried by exosomes, non-coding RNAs (ncRNAs) stand out as key regulatory molecules widely studied in the context of cancer and neurodegenerative diseases, playing vital roles in intercellular communication and immune response regulation [7,8]. Recent findings suggest that ncRNAs in exosomes also contribute to the progression of pathogen infections by modulating crucial mechanisms such as immune evasion and cell apoptosis [9,10]. For instance, in hepatocytes infected with the hepatitis B virus (HBV), the expression of exosomal miR-21 and miR-29a was significantly elevated, leading to the suppression of IL-12p35 mRNA expression and a consequent reduction in inflammatory factor secretion [11].

*L. monocytogenes*, a Gram-positive bacterium, is ubiquitously distributed in the environment and can survive intracellularly within both professional and non-professional phagocytes [12]. Recognized by the World Health Organization (WHO) as one of the four major foodborne pathogens [13], *L. monocytogenes* can lead to severe illnesses, including diarrhea, meningitis, and miscarriage, upon infection through the digestive tract in humans and animals, posing significant risks to pregnant women, the elderly, and individuals with weakened immune systems [14,15]. The bacterium’s capacity for intracellular survival and immune evasion poses substantial challenges in developing effective immune response strategies and therapeutic interventions against *L. monocytogenes* infection [16,17]. Numerous studies have highlighted the critical role of exosomes in the infection process by *L. monocytogenes*. For instance, Nandakumar et al. demonstrated that *L. monocytogenes* can manipulate host cell exosomes to transport anti-inflammatory molecules, such as IFN-β, thereby diminishing the antimicrobial responsiveness of other immune cells [18]. Similarly, Izquierdo-Serrano et al. showed that *L. monocytogenes*-infected dendritic cells release increased numbers of larger exosomes with altered protein modifications, upregulate the expression of disease-fighting genes, and exhibit protective effects during poxvirus infections [19]. Additionally, Kaletka et al. reported significant alterations in the protein and mRNA composition of trophoblast-derived extracellular vesicles (tEVs) after *L. monocytogenes* infection of placental stem cells, which heightened macrophage susceptibility to the bacterium [20]. Despite these findings, the precise mechanisms by which *L. monocytogenes* manipulates host cell exosomal ncRNAs to modulate their intracellular survival environment remain unclear [21,22].

Macrophages are central targets for *L. monocytogenes* due to their dual roles in pathogen clearance and immune modulation. While prior studies have implicated exosomes in bacterial infections, the mechanisms by which *L. monocytogenes* exploits EXO-LM ncRNAs to subvert host immunity remain poorly understood. Here, we systematically investigated the immunomodulatory function of EXO-LM through transcriptomic profiling and functional validation. By integrating RNA sequencing, dual-luciferase assays, and gain-of-function experiments, we identified a novel ceRNA axis involving lncRNA Rpl13a-213 and miR-132-3p. Our results demonstrate that *L. monocytogenes* downregulates Rpl13a-213 in EXO-LM to relieve miR-132-3p suppression, thereby dampening macrophage inflammatory responses and enhancing bacterial survival. This study provides the first evidence of a pathogen-driven exosomal ceRNA network that facilitates intracellular persistence, offering new targets for therapeutic intervention against listeriosis.

## 2. Materials and Methods

### 2.1. Exosome-Free Serum Preparation

To ensure that all exosomes in the collected cell supernatants originated exclusively from macrophages, exosome-free serum (ThermoFisher, Waltham, MA, USA) was used in the preparation of the cell culture medium. In brief, approximately 45 mL of fetal bovine serum (FBS) was transferred into an ultracentrifuge tube, equilibrated, and securely sealed. The tube was then subjected to ultracentrifugation at 160,000× *g* for 16 h at 4 °C. After centrifugation, the supernatant, representing the exosome-free serum with all exosomes thoroughly removed, was carefully collected for subsequent use.

### 2.2. Cell, Plasmid, Strain, and Culture Conditions

RAW264.7 mouse macrophages (ATCC TIB-71) were purchased from the Henan Industrial Microbial Strain Engineering Research Center. The HEK293T cells (ATCC CRL-3216), noted for their high transfection efficiency, were utilized for dual-luciferase assays and were maintained at the Key Laboratory of Preventive Veterinary Medicine, Shihezi University. RAW264.7 mouse macrophages were cultured in a DMEM cell culture medium supplemented with 10% FBS (free of exosomes), while PHK-293T cells were cultured in a 1640 medium with 10% FBS. The plasmids psiCHECK2 and pcDNA 3.1 were maintained at the Key Laboratory of Preventive Veterinary Medicine, Shihezi University, while the *L. monocytogenes* EGD-e strain was obtained from the China Microbial Strain Centre. *L. monocytogenes* was cultured in BHI broth (pH 7.2 ± 0.2) at 37 °C under aerobic conditions with shaking at 200 rpm, and **E. coli** DH5α was grown in LB broth (pH 7.0 ± 0.2) at 37 °C under aerobic conditions with shaking at 200 rpm.

### 2.3. Extraction of Exosomes from L. Monocytogenes-Infected Macrophages

RAW264.7 macrophages were infected with the *L. monocytogenes* EGD-e strain at a multiplicity of infection (MOI) of 50 for 1 h. An equivalent volume of PBS was used as a control in place of the bacterial solution. To eliminate extracellular *L. monocytogenes* bacteria, gentamicin (25 μg/mL) was added to the culture medium and incubated for 1 h, followed by replacing the medium with DMEM containing 10% FBS (no exosomes). Extracellular bacteria elimination was confirmed by plating a supernatant on BHI agar (no colonies observed after 24 h). After 24 h of LM infection, the cell culture supernatant was collected and filtered through a 15 mL 100 kDa molecular weight cutoff (MWCO) ultrafiltration membrane (Millipore, Burlington, MA, USA). The filtrate was isolated via differential ultracentrifugation at 160,000× *g* for 2 h at 4 °C, and the resulting precipitate was collected to isolate exosomes. This precipitation was resuspended to 1 mL of PBS. Exosomes were further purified by size exclusion chromatography (Exosuper purification kit, ECHO BIOTECH, Beijing, China), yielding 3.5 mL of fractions containing exosomes in each purification tube. These fractions were subsequently concentrated to 400 μL using a 2 mL 100 kDa molecular weight cutoff (MWCO) ultrafiltration membrane (Millipore). Exosomes from uninfected macrophages (hereafter EXO) were extracted using the same method.

### 2.4. Identification of Exosomes

The extracted exosomes were identified both morphologically and immunologically, as described in the literature [2,23]. In brief, 10 μL of exosomes were placed on a copper mesh, dried, and then restained with 20 g/L of phosphotungstic acid. They were observed for morphological features using transmission electron microscopy. After diluting the exosomes to the appropriate multiplicity, a nanoparticle tracking analysis (NTA) assay was conducted to determine their particle size and concentration. These samples underwent SDS-PAGE electrophoresis and Western blot (WB) analysis using fluorescently labeled mouse antibodies, specifically targeting CD9, CD63, TSG101, and Calnexin proteins, to detect the specific markers of the exosomes.

### 2.5. Effect of Exosomes on Macrophage Viability

Cell viability was assessed using the CCK-8 assay. In brief, RAW264.7 macrophages were seeded into 96-well plates at a density of approximately 5 × 10^3^ cells per well. These macrophages were stimulated with 10 μL of EXO, EXO-LM, and PBS for a duration of 12 h, and then infected with the *L. monocytogenes* EGD-e strain at a multiplicity of infection (MOI) of 100 for 1 h. Following this, the cell culture medium was replaced with one containing gentamicin (25 μg/mL) to eliminate extracellular *L. monocytogenes*. CCK-8 assays were conducted 8, 16, 24, 32, 40, and 48 h post-infection. In the experiment, six parallel assays were conducted for each sample, wherein 10 μL of CCK-8 solution was added to each well, followed by an incubation period of 1 h at 37 °C. After incubation, absorbance was measured at 450 nm using an enzyme-linked immunosorbent assay (ELISA) reader.

### 2.6. Impact of Exosomes on Cytokine Expression in Macrophages

A double-antibody sandwich ELISA kit was employed to measure cytokine expression levels following exosome stimulation of RAW264.7 macrophages. Briefly, cell supernatants from RAW264.7 macrophages stimulated with EXO and EXO-LM for 24 h were collected and centrifuged at 3000 rpm for 10 min to remove particles and aggregates. Subsequently, 10 of the sample, 40 μL of sample diluent, and 100 μL of horseradish peroxidase (HRP) conjugate were added to each well. The wells were then sealed with a plate-sealing membrane and incubated at 37 °C for 1 h. Following incubation, the liquid was discarded, the wells were blotted dry using absorbent paper, and the plate was washed five times with washing solution. Next, 50 μL each of substrate solutions A and B was added to each well, and the plate was incubated at 37 °C for 15 min while being protected from light. Finally, 50 μL of stop solution was added, and the absorbance at 450 nm was measured within 15 min.

### 2.7. Real-Time Fluorescence Quantitative PCR

The cells were washed three times with PBS and lysed with TRIzol™ reagent, followed by the extraction of the total RNA. The RT-qPCR primers were designed using SnapGene (version 7.1.2) software, based on the mouse gene sequences from GenBank(GCA_000001635.9). Using the RNA concentration as a guide, an appropriate amount of total RNA was reversely transcribed into cDNA with the PrimeScript™ RT-PCR Kit (Takara, Kusatsu, Shiga, Japan). Then, the RT-qPCR assay was conducted with PerfectStart^®^ Green qPCR Super Mix (TransGen Biotech, Beijing, China). Subsequently, a two-step method was employed to calculate the relative expression of cytokine genes such as *IL-1β*, *IL-4*, *IL-6*, *IL-10*, *TNF-α*, and *INF-γ* (Table 1).

### 2.8. Transcriptome Sequencing of Macrophages After Exosome Stimulation

Macrophages were stimulated with 10 μL of EXO and EXO-LM for 24 h. Afterward, cells were lysed using TRIzol™ reagent, and total cellular RNA was extracted. The extracted RNA was processed with the Hieff NGS Ultima Dual-Mode mRNA Library Prep Kit for Illumina (Yeasen Biotechnology, Shanghai Co., Ltd., Shanghai, China) to create sequencing libraries. These libraries were then sequenced on the Illumina NovaSeq platform, producing 150 bp paired-end reads (average depth: 30× per sample). Three independent biological replicates were analyzed for each group. The sequencing data were analyzed using Hisat2 (version 2.0.4) and StringTie (version 2.2.1) software, which facilitated alignment and assembly with the reference genome. Novel genes were identified using DIAMOND (version 2.0.15) software. Differential expression analysis was performed with EdgeR (version 3.32.1), applying the criteria of Fold Change ≥ 2 and FDR < 0.05 for significance. Gene enrichment analysis was conducted using the R package *clusterProfiler* (version 4.4.4). Functional annotations for the enriched genes were derived from the KEGG Ortholog database (http://www.genome.jp/kegg/, 16 August 2023) and Gene Ontology datasets (http://www.geneontology.org/ 16 August 2023). Additionally, macrophage miRNA transcriptome libraries were constructed as described in the literature and sequenced following the same methodology.

### 2.9. Exosomal RNA Extraction and Sequencing

Total exosomal RNA was extracted using the exoRNeasy Midi Kit (Qiagen, Hilden, Germany), following the manufacturer’s protocol. For each sample, 1 μg of total RNA was employed as the starting material to construct long-chain RNA libraries. Strand-specific libraries were prepared using the NEBNext^®^ Ultra™ RNA Library Prep Kit for Illumina^®^ (New England Biolabs, Ipswich, MA, USA), adhering to the manufacturer’s instructions. Each sample was assigned a unique index tag to enable differentiation of sequence features, and sequencing was conducted as described previously. Three independent biological replicates were analyzed for each group. Differential expression analysis was performed using EdgeR for lncRNA and mRNA and DESeq2 for circRNA. Transcript and gene expression levels were normalized using FPKM. Screening criteria were defined as a fold change ≥ 2 and an adjusted *p*-value (padj) < 0.05.

### 2.10. Dual-Luciferase Reporter System

The target binding site of Rpl13a-213 to miR-132-3p was predicted using a bioinformatics database. Wild-type and mutant sequences (Rpl13a-213, Rpl13a-213-A, Rpl13a-213-B, and Rpl13a-213-AB) were designed and cloned into the psiCHECK2 plasmid vector. HEK293T cells were then seeded in 12-well plates, and when cell confluence reached 70–80%, the cells were co-transfected with the vectors psiCHECK2-mir132-WT, psiCHECK2-mir132-MUTA, psiCHECK2-mir132-MUTB, or psiCHECK2-mir132-MUTAB. Following this, the cells were treated with either a miR-186-5p mimic (Shanghai Bioengineering, Shanghai, China) or a negative control (NC-mimic) (Shanghai Bioengineering) for 48 h. Finally, luciferase activity was assessed using a dual-luciferase reporter assay system (Allotype Gold).

### 2.11. Cell Transfection and Cellular miRNA Detection

The Rpl13a-213 overexpression plasmid was constructed by recombining the DNA fragment of the *Rpl13a-213* gene from the genome with the pcDNA 3.1 plasmid, synthesized by Shanghai Bioengineering Co. Transfection of the Rpl13a-213 overexpression plasmid into RAW264.4 macrophages was carried out using the Lipofectamine™ 3000 Transfection Kit (ThermoFisher, Waltham, MA, USA). miRNA was reverse-transcribed with the miRNA 1st Strand cDNA Synthesis Kit (utilizing the stem-loop method). miRNA-specific primers were designed using the Novozymes miRNA Primer Design (version 1.01) Software. The reverse transcription primer sequence used was as follows: 5′-GTCGTATCCAGTGCAGGGTCCGAGGTATTCGCACTGGATACGACC. RT-qPCR primers are listed in Table 1.

### 2.12. Statistical Analysis of Data

All data were derived from three independent experiments and are presented as mean ± SD. Statistical analyses were conducted using one-way or two-way ANOVA through GraphPad Prism 9.5 software. Significance was determined at *p* < 0.05, with results considered highly significant at *p* < 0.01. 

## 3. Result: Subsection

### 3.1. Macrophage-Derived Exosomes Isolation and Characterization

Exosomes were successfully isolated using differential ultracentrifugation and size exclusion chromatography. Transmission electron microscopy (TEM) micrographs revealed that exosomes derived from both *L. monocytogenes*-infected macrophages (EXO-LM) and uninfected macrophages (EXO) measured approximately 60 to 80 nm in diameter. These exosomes displayed round or near-round vesicular structures with a double-layer membrane, which aligns with typical exosome characteristics (Figure 1A). Nanoparticle tracking analysis further demonstrated that the particle size distribution ranged from 60 to 100 nm, with a single-peak profile. Moreover, the number of exosome particles in the EXO-LM group was notably higher than that in the exosomes group (Figure 1B). Western blot analysis validated the presence of exosomal marker molecules CD9, CD63, and TSG101 on the surface of the extracted Exo stocks (Figure 1C). Original uncut western blots in Appendix A. Results from the nanoflow assay confirmed a significantly higher count of the surface protein-positive particles, such as CD9, CD63, and TSG101, in both the EXO and EXO-LM groups compared to the supernatants of unisolated cells. Additionally, the EXO-LM group exhibited a greater number of positive particles compared to the EXO group (Figure 1D). 

### 3.2. Immunosuppressive Effects of EXO-LM

To investigate the impact of EXO-LM on the immune system, we analyzed the secretion of inflammatory factors in macrophages following stimulation with EXO-LM. As illustrated, in comparison to the group stimulated by EXO alone, macrophages in the EXO-LM-stimulated group exhibited significant polarization (Figure 2A). At 24 h, both M1-type macrophages (polygonal shape with short pseudopods extending from the periphery) and M2-type macrophages (elongated spindle shape with long pseudopods extending from the ends) were observed, and M2-type macrophages were further increased after 48 h. Similarly, compared to the EXO-stimulated group, the expression of pro-inflammatory cytokines *IL-1β*, *IL-6*, and *TNF-α* in the EXO-LM stimulated group reached a peak at 12 h and subsequently declined over time, whereas the expression of the anti-inflammatory cytokine *IL-10* continued to increase (Figure 2B). These findings were further corroborated by the cytokine ELISA assay, which demonstrated consistency with the RT-qPCR results (Figure 2C).

The CCK-8 assay results revealed that cell viability in the EXO-LM-stimulated group was significantly higher than that in the EXO-stimulated group, showing a steady upward trend over time (Figure 2D). Moreover, data from the colony-forming unit (CFU) assay indicated that macrophages pre-stimulated with EXO-LM significantly suppressed bacterial survival (Figure 2E). However, bacterial survival began to increase after 48 h. These findings suggest that may contribute to maintaining bacterial survival by dampening the inflammatory response of macrophages.

### 3.3. EXO-LM Alters the Macrophage Transcriptome

To further elucidate the impact of EXO-LM on the biological functions of macrophages, high-throughput transcriptome sequencing technology was employed to perform an in-depth analysis of macrophages following EXO-LM stimulation. As illustrated in Figure 3A, the expression levels of 532 genes were significantly upregulated while 336 genes were markedly downregulated in macrophages after stimulation with EXO-LM (Figure 3A). Notably, the expression patterns of inflammatory factor genes, including *IL-1β*, *IL-10*, and *TNF-α*, were consistent with the results obtained through RT-qPCR. Furthermore, 109 differentially expressed miRNAs were identified, among which 65 were upregulated and 44 were downregulated, as shown in Figure 3B.

The Gene Ontology (GO) analysis revealed that EXO-LM stimulation significantly influenced various biological processes in macrophages, including those related to the immune system, signal transduction, metabolism, genetic information processing, cell growth and death, and the endocrine system. KEGG and GSEA analyses further demonstrated that the transcriptional levels of key genes, such as inhibitors of *IKB-α* and *MKP5*, along with those central to inflammatory pathways (e.g., NF-kappa-B and MAPK), were markedly upregulated (Figure 3C,D). Conversely, the expression of cell cycles regulatory genes, including *Cyclin E-CDK2*, *Cyclin A-CDK2*, and *Cyclin B-CDK1*, as well as the critical enzymes involved in the DNA replication process (e.g., *Mcm4*, *Mcm5*, *Mcm6*, *Mcm7*), was significantly downregulated at the transcriptional level. Additionally, genes promoting apoptosis, such as *Bax*, *DIABLO*, and *BIRC5*, were substantially downregulated, while *Bcl-2*, a gene associated with apoptosis inhibition, exhibited significant upregulation. These transcriptome analysis results suggest that EXO-LM disrupted the biological cycle of macrophages, suppressing inflammatory and apoptotic responses and thereby creating a favorable microenvironment for *L. monocytogenes* survival within macrophages.

### 3.4. EXO-LM Transcriptome Sequencing Analysis

To elucidate the molecular mechanisms of EXO-LM in immune regulation, transcriptome sequencing was employed to analyze the differential RNA expression of macrophage-derived EXO-LM. As depicted in Figure 4A, 295 differentially expressed mRNAs were identified, with 209 showing upregulation and 86 showing downregulation (Figure 4A). Notably, the MAPK signaling pathway was significantly downregulated, as revealed by KEGG and GSEA analyses (Figure 4B,C). For differential expression analysis of non-coding RNAs (ncRNAs), 81 differentially expressed lncRNAs were identified, with 32 upregulated and 49 downregulated (Figure 4D). Regarding circRNAs, 51 were identified, with 5 upregulated and 46 downregulated; among these, 26 contained miRNA binding sites (Figure 4E). For miRNAs, a total of nine were identified, with one upregulated and eight downregulated (Figure 4F).

Using the transcriptome sequencing alongside bioinformatics tools such as TargetScan, ENCORI, and regRNA 2.0, the potential target genes of differentially expressed lncRNAs, circRNAs, and miRNAs were predicted. A competing endogenous RNA (ceRNA) regulatory network was constructed, consisting of exosomal lncRNAs, circRNAs, miRNAs, and their target macrophage miRNAs and mRNAs. Comparative analyses identified a potential regulatory interaction between the key regulatory molecules lncRNA Rpl13a-213 and miR-132-3p, indicating that this molecular pair plays a crucial role in macrophage immune function (Figure 4G).

### 3.5. Regulation of Macrophage Function by EXO-LM Through Rpl13a-213

To unveil the specific mechanism of interaction between Rpl13a-213 and miR-132-3p in EXO-LM, this study utilized predictive analyses from bioinformatics tools such as miRanda, PITA, and RNAhybrid. The analysis identified two potential binding sites between lncRNA Rpl13a-213 and miR-132-3p. Overexpression of Rpl13a-213 reduced luciferase activity by 62.3% ± 5.1% compared to the mutant control (*p* < 0.01, Figure 5A). These binding sites were experimentally validated using the dual-luciferase reporter assay (Figure 5B). To assess the impact of Rpl13a-213 and miR-132-3p on macrophage function, macrophages stimulated with EXO-LM were transfected with these two molecules and incubated for 24 h. Changes in cytokine expression were subsequently measured. The results demonstrated that overexpression of Rpl13a-213 significantly upregulated the expression of pro-inflammatory cytokines in macrophages compared to the control group, whereas overexpression of miR-132-3p exerted the opposite effect (Figure 5C). Further exploration of the role of these two molecules in *L. monocytogenes* viability revealed that macrophages overexpressing Rpl13a-213 effectively inhibited bacterial survival following *L. monocytogenes* infection. In contrast, macrophages pre-treated with miR-132-3p displayed a marked increase in bacterial survival (Figure 5D).

## 4. Discussion

Exosomes represent a novel cell-to-cell communication system that significantly influences the physiological state of cells, which is intricately linked to the onset and progression of various diseases [24,25]. In recent years, there has been an increasing focus on the role of Exo in intercellular communication, particularly the ncRNAs they carry. These ncRNAs act as crucial regulatory molecules, playing significant roles in pathogen infections and immune regulation [26]. Wang et al. demonstrated that there was a marked increase in miR-155 expression within macrophage-derived Exo during *Helicobacter pylori* infection, which subsequently caused a significant reduction in inflammatory signaling-pathway-associated proteins, such as MyD88 and NF-κB, in target cells [27]. To date, the intracellular survival and immune evasion strategies of *L. monocytogenes*, a pathogen associated with severe foodborne illnesses, have not been fully elucidated. However, exosomes appear to play a critical role in these mechanisms [28,29]. Consequently, this study aims to investigate the regulatory function of EXO-LM released by macrophages in the context of *L. monocytogenes* infection. A key focus is placed on examining changes in the ncRNAs contained within EXO-LM to unravel the molecular mechanisms by which exosomal ncRNAs facilitate the survival of *L. monocytogenes*.

Here, we investigated the effects of EXO-LM on macrophage-based cells through cell infection experiments. The results demonstrated that EXO-LM regulates macrophage polarization, the expression of inflammatory factors, and the apoptotic process, thereby creating conditions conducive to *L. monocytogenes* infection and survival. Notably, an inflammatory suppression response and differentiation into M2-type macrophages were observed 24 h after EXO-LM stimulation, whereas similar changes emerged only on the third day following *L. monocytogenes* infection in mice [22]. The expression of pro-inflammatory cytokines IL-1β, IL-6, and TNF-α in EXO-LM-stimulated macrophages peaked during the early stages but gradually declined over time, while the expression of the anti-inflammatory cytokine IL-10 showed a consistent upward trend. Additionally, we observed that macrophages stimulated by EXO-LM exhibited significantly enhanced cell viability and reduced apoptosis following *L. monocytogenes* infection, particularly during the later stages of infection. These cells also strongly facilitated bacterial survival.

Transcriptomic analyses revealed that EXO-LM profoundly influences several biological processes, including immune response, signaling, and macrophage metabolism. KEGG analyses further demonstrated that EXO-LM disrupts critical signaling pathways involved in immune responses, such as NF-kappa B, MAPK, and TNF. Moreover, it suppresses pathways related to the cell cycle and apoptosis, fostering the formation of a microenvironment favorable for bacterial survival. These findings suggest that *L. monocytogenes* can exploit host cell exosomes to deliver diverse signaling molecules, thereby diminishing the antimicrobial capacity of macrophages. This mechanism allows *L. monocytogenes* to effectively manipulate host immune responses to facilitate its survival. Consistent with previous findings by Nandakumar et al. [18] and Kaletka et al. [20], *L. monocytogenes* have been shown to manipulate host-derived exosomes as a strategy to subvert immune surveillance, further corroborating the critical role of extracellular vesicle-mediated mechanisms in bacterial immune evasion.

To elucidate the molecular mechanisms of EXO-LM in immune regulation, this study employed whole-transcriptome sequencing to analyze RNA alterations in macrophage-derived EXO-LM following *L. monocytogenes* infection. Transcriptomic analysis revealed numerous differentially expressed mRNAs, lncRNAs, circRNAs, and miRNAs in EXO-LM. Functional predictions suggested that these ncRNAs may regulate various immune-related genes and signaling pathways by modulating their target gene expression. Based on these findings, this study constructed a ceRNA regulatory network between EXO-LM and macrophages, identifying a potential interaction between the exosomal lncRNA Rpl13a-213 whose expression was significantly reduced and the macrophage miR-132-3p, which exhibited markedly increased expression [30,31]. Dual-luciferase reporter assays further validated that lncRNA Rpl13a-213 exerts a regulatory effect on miR-132-3p. Previous studies have shown that miR-132-3p directly or indirectly targets key immune-related genes, including *NF-κB*, *p65*, and *IκBα*, playing a crucial role in suppressing the inflammatory response of macrophages [32,33,34]. Based on these observations, the study hypothesized that *L. monocytogenes* infection regulates immune responses through the interaction of Rpl13a-213 and miR-132-3p (Figure 6). To confirm this hypothesis, an Rpl13a-213 overexpression plasmid was constructed. Overexpression of Rpl13a-213 was found to significantly reduce miR-132-3p expression in EXO-LM-treated macrophages while simultaneously promoting the expression of pro-inflammatory cytokines such as IL-1β and IL-6. Taking together, these findings demonstrate that lncRNA Rpl13a-213 modulates the inflammatory response of macrophages by inhibiting miR-132-3p, thereby influencing the intracellular survival of *L. monocytogenes*.

In conclusion, our study demonstrates the regulatory function of exosomes secreted by *L. monocytogenes*-infected macrophages on host immune cells, which confirms the involvement of exosome ncRNAs in pathogen infection and the host immune response. These findings offer a new perspective on the molecular mechanisms underlying the intracellular survival and immune evasion of *L. monocytogenes*.

## Figures and Tables

**Figure 1 microorganisms-13-00410-f001:**
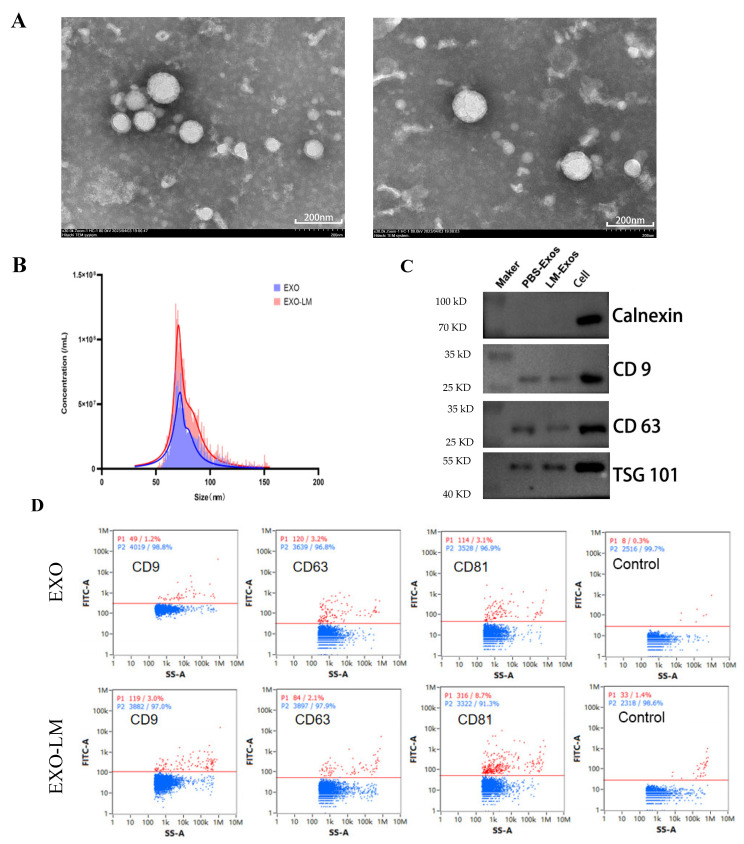
Extraction of exosomes from macrophages. (**A**) Transmission electron microscopy observations of the morphology of EXO (A1) and EXO-LM (A2). (**B**) Measurement of particle size of EXO and EXO-LM by nanoparticle tracking analysis (NTA). (**C**) Analysis of exosome markers (CD9, CD63, and TSG101) on the surface of EXO and EXO-LM by Western blot. (**D**) Identification of exosome markers (CD9, CD63, and CD81) on the surface of EXO and EXO-LM by nanoflowometry.

**Figure 2 microorganisms-13-00410-f002:**
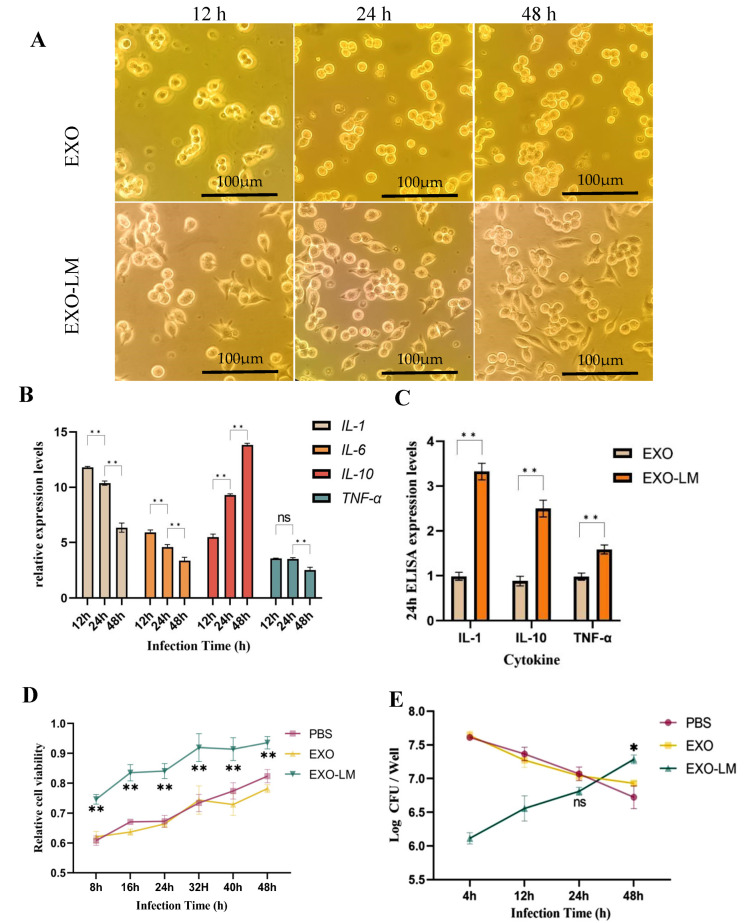
The effects of EXO/EXO-LM on macrophages. (**A**) The morphology of macrophages treated by EXO and EXO-LM. (**B**) Detection of the relative expression levels of inflammation-related cytokine mRNA in macrophages treated by EXO and EXO-LM. (**C**) Analysis of the expression of inflammation-related cytokines in macrophages treated by EXO/EXO-LM. (**D**) Cell viability assay of macrophage treated by EXO/EXO-LM. (**E**) Determination of intracellular survival of LM in macrophages treated by EXO/EXO-LM. Note: data are represented as the mean ± SD of three independent experiments. Significant differences are indicated by * *p* < 0.05 and ** *p* < 0.01, respectively. Non-significant differences are indicated by ns.

**Figure 3 microorganisms-13-00410-f003:**
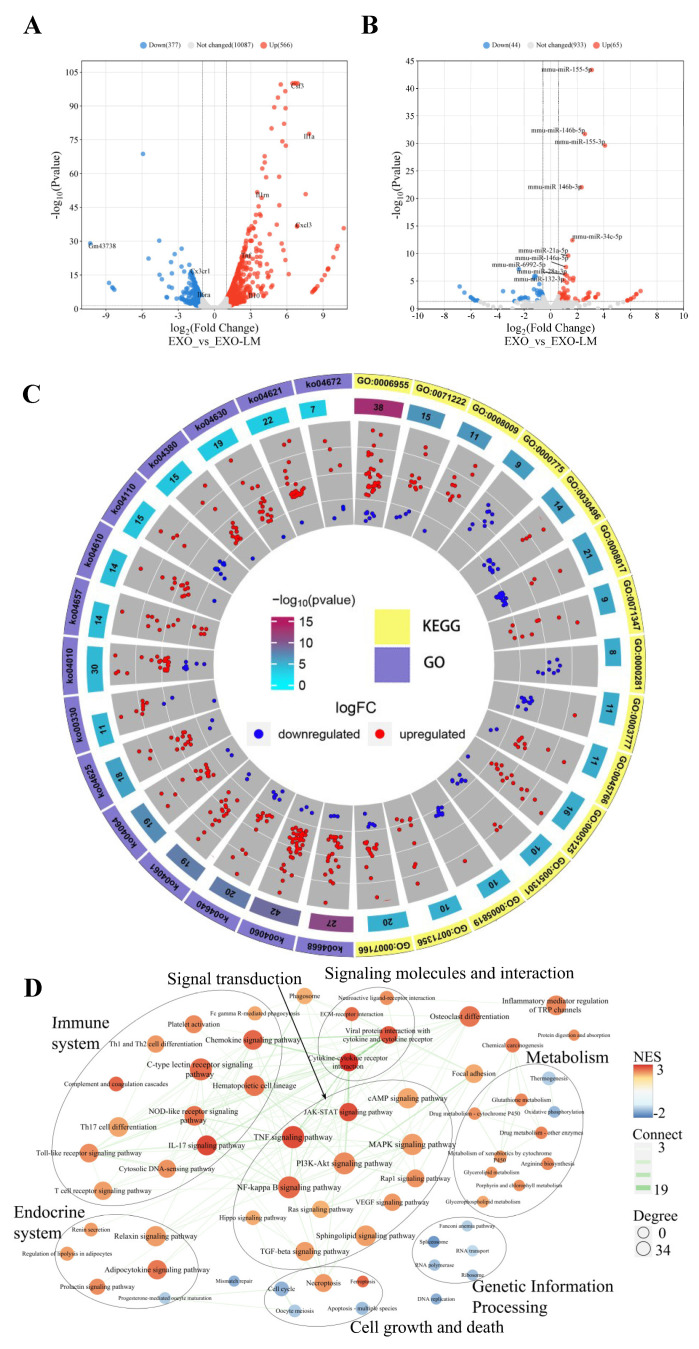
Transcriptomic analysis of macrophages stimulated with EXO/EXO-LM. (**A**,**B**) Volcano plots of differentially expressed mRNAs and miRNAs in macrophages stimulated with EXO and EXO-LM, respectively. (**C**) Circle plots of GO and KEGG pathway enrichment analysis for differentially expressed genes (the top 15 pathways with the smallest *p*-values). (**D**) Construction of interaction networks using Gene Set Enrichment Analysis (GSEA) based on KEGG pathways as gene sets.

**Figure 4 microorganisms-13-00410-f004:**
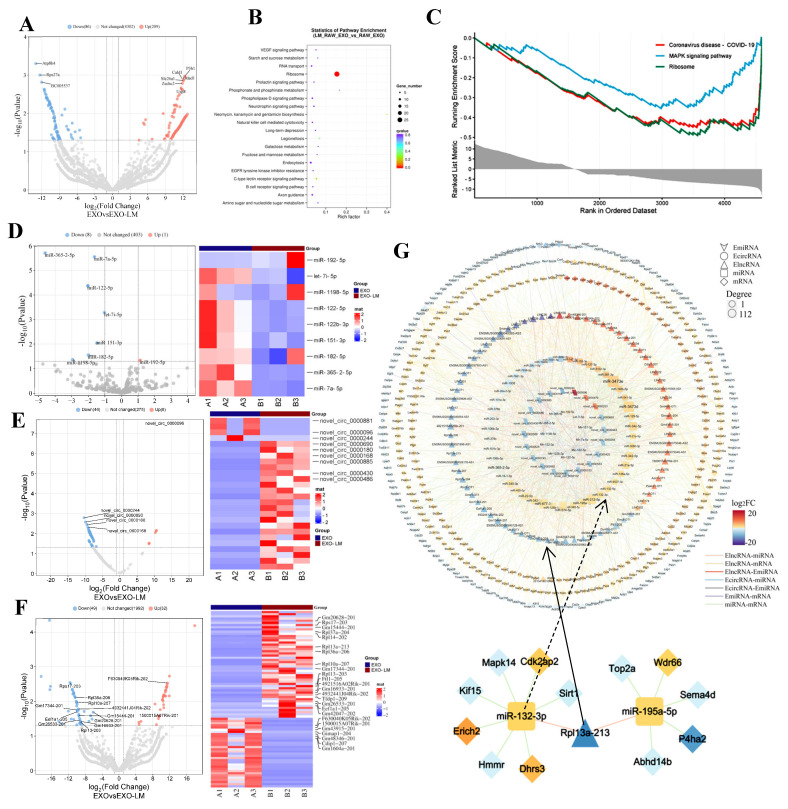
Transcriptome-wide analysis of EXO and EXO-LM from macrophages. (**A**–**C**): Volcano plots of differentially expressed genes between EXO and EXO-LM, and KEGG and GSEA enrichment analyses. (**D**) Volcano plots and heatmaps of differentially expressed miRNA genes between EXO and EXO-LM. (**E**) Volcano plots and heatmaps of differentially expressed circRNA genes between EXO and EXO-LM. (**F**) Volcano plots and heatmaps of differentially expressed lncRNA genes between EXO and EXO-LM. (**G**) Combined with the results predicted by miRNAda (http://www.microrna.org, 29 August 2023), PITA (http://mirtoolsgallery.tech/mirtoolsgallery/node/1066, 3 September 2023), and RNAhybrid(http://bibiserv.techfak.uni-bielefeld.de/rnahybrid/, 3 September 2023) software, a regulatory diagram of differentially expressed ElncRNA, EcircRNA, EmiRNA, and target macrophage miRNA, mRNA between EXO and EXO-LM was constructed utilizing Cytoscape(version 3.10.1) software. Volcano plots show differentially expressed RNAs (red: upregulated; blue: downregulated; dashed lines: fold change ≥ 2, FDR < 0.05). Heatmaps display hierarchical clustering of top 50 genes (color scale: log2 fold change).

**Figure 5 microorganisms-13-00410-f005:**
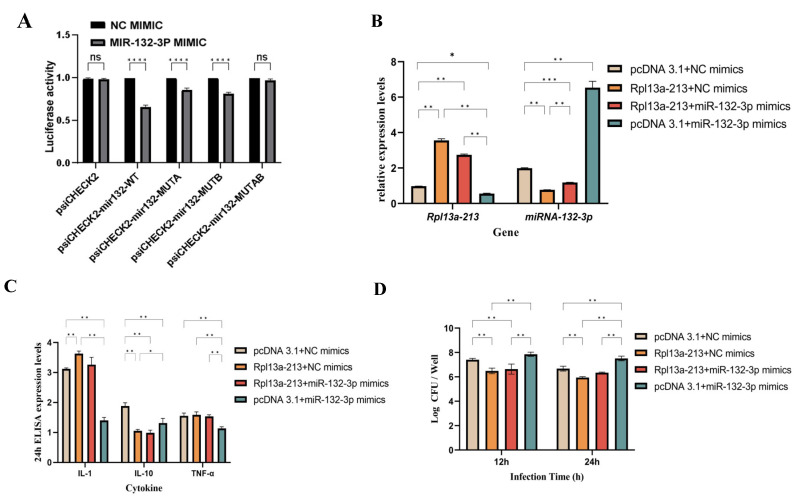
The immunomodulatory roles of Rpl13a-213-mir-13-3p in EXO-LM from macrophages. (**A**) Verification of the targeting relationship between Rpl13a-213 and mir-32-3p by dual-luciferase reporter systems. (**B**) Analysis of Rpl13a-213 and mir-12-3p expression in macrophages pre-treated with EXO-LM by RT-qPCR. (**C**) Determination of cytokine expression in macrophages pre-treated with EXO-LM. (**D**): Determination of intracellular survival of LM in macrophages treated with EXO/EXO-LM. Note: data are expressed as ± SD of three independent experiments; significant differences are indicated by * *p* < 0.05, ** *p* < 0.01, *** *p* < 0.001, **** *p* < 0.0001. Non-significant differences are indicated by ns.

**Figure 6 microorganisms-13-00410-f006:**
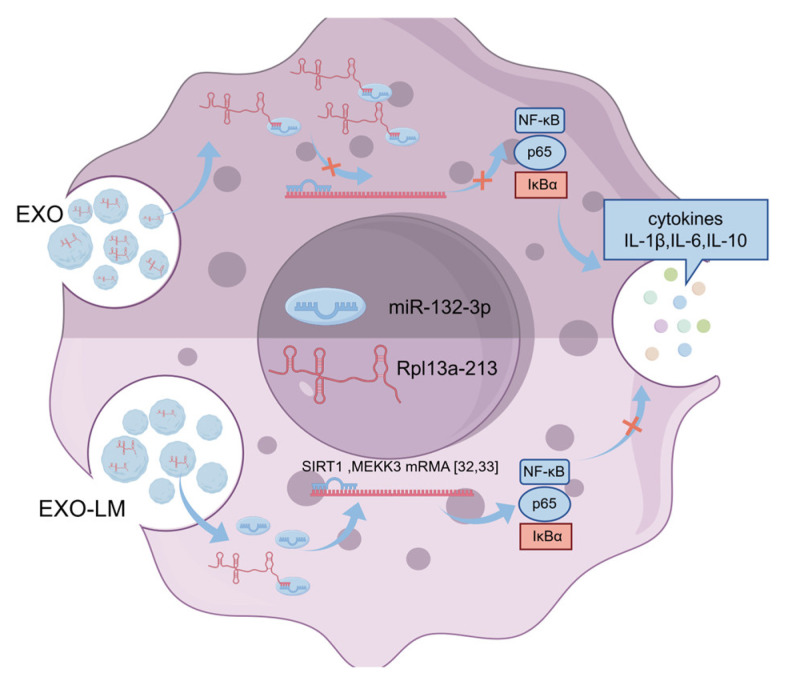
Diagram of the regulatory mechanism of Rpl13a-213 and mir-12-3p. Red x represents regulatory interference.

**Table 1 microorganisms-13-00410-t001:** List of primer sequences used in this study.

Target Gene	Primer Name	Primer Sequence (5′→3′)
*IL-1β*	F1	GCAACTGTTCCTGAACTCAACT
	R1	ATCTTTTGGGGTCCGTCAACT
*IL-4*	F2	ACAGGAGAAGGGACGCCAT
	R2	GAAGCCCTACAGACGAGCTCA
*IL-6*	F3	TCCAGTTGCCTTCTTGGGACTG
	R3	TTGGATGGTCTTGGTCCTTAGCC
*IL-10*	F4	GCTCTTACTGACTGGCATGAG
	R4	CGCAGCTCTAGGAGCATGTG
*TNF-α*	F5	TCCAGTTGCCTTCTTGGGACTG
	R5	TTGGATGGTCTTGGTCCTTAGCC
*INF-γ*	F6	ACAGGAGAAGGGACGCCAT
	R6	GAAGCCCTACAGACGAGCTCA
Rpl13a-213-WT	F7	TACCATCTGAGGCTGTTCTAGCCCTT
	R7	GAGGCAAACAGTCTTTATTGGGTTCACAC
Rpl13a-213-MUTA	F8	TACCATCATTGGCTGTTCTAGCCCTTG
	R8	GAGGCAAACAGTCTTTATTGGGTTCACAC
Rpl13a-213-MUTB	F9	TACCATCTGAGGCTGTTCTAGCCCTT’
	R9	GAGGCAAAGCAAGTTTATTGGGTTCACAC
Rpl13a-213-MUTAB	F10	TACCATCATTGGCTGTTCTAGCCCTTG
	R10	GAGGCAAAGCAAGTTTATTGGGTTCACAC
psiCHECK2-mir132-WT	F11	CCGCTCGAGTACCATCTGAGGCTGTTCTAG
	R11	ATAAGAATGCGGCCGCGAGGCAAACAGTCTTTATTGG
psiCHECK2-mir132-MUTA	F12	CCGCTCGAGTACCATCTGAGGAGTTTCTAG
	R12	ATAAGAATGCGGCCGCGAGGCAAACAGTCTTTATTGG
psiCHECK2-mir132-MUTB	F13	CCGCTCGAGTACCATCTGAGGCTGTTCTAG
	R13	ATAAGAATGCGGCCGCGAGGCAAAGCAAGTTTATTGG
psiCHECK2-mir132-MUT	F14	CCGCTCGAGTACCATCTGAGGAGTTTCTAG
	R14	ATAAGAATGCGGCCGCGAGGCAAAGCAAGTTTATTGG

## Data Availability

The datasets presented in this article are not readily available because the data are part of an ongoing study. Requests to access the datasets should be directed to qj710625@shzu.edu.cn.

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
