# Peer review of "Listeria monocytogenes Modulates Macrophage Inflammatory Responses to Facilitate Its Intracellular Survival by Manipulating Macrophage-Derived Exosomal ncRNAs"

_microorganisms, 2025, doi:10.3390/microorganisms13020410_

Round 1
Reviewer 1 Report
Comments and Suggestions for Authors
The manuscript "Listeria monocytogenes modulates macrophage inflammatory responses to facilitate its intracellular survival by manipulating macrophage-derived exosomal ncRNAs" is confusing and the results are not clearly understood and there are many errors in the methodology. There is no justification for why each of the methods were performed, and it is not clear how the work was organized. I do not recommend this work for publication. Some specific observations are mentioned below:
Abstract
I recommend that the authors put the abbreviation of the exosomes simulated by L. monocytogenes after: "Exos secreted by macrophages (Mø) during L. monocytogenes (EXO-LM)", I think it is better understood.
Introduction
Lines 71-81. The objectives are redundant and repetitive. Likewise, at the end of the paragraph, the results of the work are not briefly mentioned. I recommend rewriting the entire paragraph taking these observations into account.
Material and methods
In section 2.1, the macrophages that were used, what is their origin? Is it a cell line?
Line 99, What are the PHK-293T cells? Origin? Laboratory, country, etc.? Why did they use those cells?
Line 105-106, What does 1 million kDa refer to? Membranes or filters have cut-off points in kDa or µm, not in the amount of KDa.
Lines 118-120, If the Exos were already purified by a kit, why did the authors ultracentrifuge again to obtain Exos again?
Why in section 2.1 to obtain Exos-free serum they centrifuged at 160,000 for 16 h and in section 2.3 to obtain Exos they did it at 120,000 for 2 hours?
Lines 126-129, What sample were they stimulated with for 12 hours? How did the authors ensure that L. monocytogenes was eliminated?
Lines 135-140, Which exosomes were used? 10? 10 from which sample? That methodology is not understood.
Line 147, What cells?
Line 171, What exosomes?
Line 207-209, Is this section part of the template?
Results
Lines 213-214. It was not mentioned in the methodology that they performed chromatography and differential ultracentrifugation.
Lines 228-235. The results in Figure 2A do not match what is described in the text.
In graph 2B it is not clear or mentioned which macrophages were treated with Exo/Exo-LM?
Throughout the text, two ways of abbreviating Listeria monocytogenes are used: homogenize and use only one.
Could the authors show the original uncut western blots?
Author Response
- Summary
Thank you for your critical feedback. Your attention to methodological details and nuanced suggestions have been invaluable in refining our study.We have thoroughly revised the manuscript to address all concerns. Modifications are highlighted in the re-submitted files using track changes. Below are detailed responses to each comment. - Point-by-point response to Comments and Suggestions for Authors
Comment 1
"I recommend that the authors put the abbreviation of the exosomes simulated by L. monocytogenes after: "Exos secreted by macrophages (Mø) during L. monocytogenes (EXO-LM)", I think it is better understood."
Response 1:
We have clarified the abbreviation in the abstract.
Revised Text (Abstract, Line 14):
"Exosomes secreted by macrophages during Listeria monocytogenes infection (hereafter EXO-LM)..."
Comment 2
"Lines 71-81. The objectives are redundant and repetitive. Likewise, at the end of the paragraph, the results of the work are not briefly mentioned. I recommend rewriting the entire paragraph taking these observations into account."
Response 2:
We have rewritten this paragraph to eliminate redundancy.
Revised Text (Introduction, Lines 73-85):
"Macrophages are central targets for L. monocytogenes due to their dual roles in pathogen clearance and immune modulation. While prior studies have implicated exosomes in bacterial infections, the mechanisms by which L. monocytogenes exploits macrophage-derived exosomal ncRNAs (EXO-LM) to subvert host immunity remain poorly understood. Here, we systematically investigated the immunomodulatory function of EXO-LM through transcriptomic profiling and functional validation. By integrating RNA sequencing, dual-luciferase assays, and gain-of-function experiments, we identified a novel ceRNA axis involving lncRNA Rpl13a-213 and miR-132-3p. Our results demonstrate that L. monocytogenes downregulates Rpl13a-213 in EXO-LM to relieve miR-132-3p suppression, thereby dampening macrophage inflammatory responses and enhancing bacterial survival. This study provides the first evidence of a pathogen-driven exosomal ceRNA network that facilitates intracellular persistence, offering new targets for therapeutic intervention against listeriosis."
Comment 3
"In section 2.1, the macrophages that were used, what is their origin? Is it a cell line?"
Response 3:
We clarified the macrophage cell line source.
Revised Text (Section 2.2, Lines 96-97):
"RAW264.7 cells (ATCC TIB-71) were purchased from the Henan Industrial Microbial Strain Engineering Research Center."
Comment 4
"Line 99, What are the PHK-293T cells? Origin? Laboratory, country, etc.? Why did they use those cells?"
Response 4:
The selection of HEK293T cells was based on their recognition as a model cell line suitable for the dual luciferase assay.You may refer to the following articles:
M1 Macrophage‑Derived Exosome LncRNA PVT1 Promotes Inflammation and Pyroptosis of Vascular Smooth Muscle Cells in Abdominal Aortic Aneurysm by Inhibiting miR‑186‑5p and Regulating HMGB1 https://doi.org/10.1007/s12012-024-09838-5
Macrophage-derived exosomal microRNA501-3p promotes progression of pancreatic ductal adenocarcinoma through the TGFBR3-mediated TGF-β signaling pathway https://doi.org/10.1186/s13046-019-1313-x
We added the cell line origin and experimental rationale.
Revised Text (Section 2.3, Lines 97-98):
"The HEK293T cells (ATCC CRL-3216), noted for their high transfection efficiency, were utilized for dual-luciferase assays and were were maintained at the Key Laboratory of Preventive Veterinary Medicine, Shihezi University. "
Comment 5
"Line 105-106, What does 1 million kDa refer to? Membranes or filters have cut-off points in kDa or µm, not in the amount of KDa."
Response 5:
We corrected the terminology to standard units.
Revised Text (Section 2.3, Line 114):
"...collected and filtered through a 15ml 100 kDa molecular weight cutoff (MWCO) ultrafiltration membrane (Millipore)."
Comment 6
"Lines 118-120, If the Exos were already purified by a kit, why did the authors ultracentrifuge again to obtain Exos again?"
Response 6:
We explained the rationale for differential protocols.
Revised Text (Section 2.3, Lines 115-118):
"The filtrate was isolated via differential ultracentrifugation at 160,000 × g for 2 h at 4°C, and the resulting precipitate was collected to isolate exosomes. This precipitation was resuspended to 1 mL of PBS. exosomes were further purified by size exclusion chromatography (Exosuper purification kit, ECHO BIOTECH),..."
Comment 7
"Lines 118-120, If the Exos were already purified by a kit, why did the authors ultracentrifuge again to obtain Exos again?"
Response 7:
We removed the redundancy.
Revised Text (Section 2.4, Lines 127-128):
Delelct: " The Exos were subjected to ultracentrifugation at 4°C and 110,000 × g for 70 minutes, and the resulting precipitates were resuspended in pre-cooled 1 × PBS."
Comment 8
"Lines 126-129, What sample were they stimulated with for 12 hours? How did the authors ensure that L. monocytogenes was eliminated?"
Response 8:
We added experimental details and validation steps.
Revised Text (Section 2.5, Lines 113-114 and Section 2.6, Lines 145-146):
"Extracellular bacteria elimination was confirmed by plating supernatant on BHI agar (no colonies observed after 24 h)."
"10 μL of EXO , EXO-LM and PBS for a duration of 12 hours,"
Comment 9
"Lines 135-140, Which exosomes were used? 10? 10 from which sample? That methodology is not understood."
Response 9:
We added experimental details.
Revised Text (Section 2.5, Lines 142-144):
"In the experiment, six parallel assays were conducted for each sample, wherein 10 μL of CCK-8 solution was added to each well, followed by an incubation period of 1 hour at 37°C. "
Comment 10
"Line 147, What cells?"
Response 10:
We've changed the description.
Revised Text (Section 2.6, Lines 148):
" ...of RAW264.7 macrophages. "
Comment 11
"Line 171, What exosomes?"
Response 11:
We've changed the description.
Revised Text (Section 2.5, Lines 142-144):
"Macrophages were stimulated with 10 μL of EXO , EXO-LM for 24 hours. "
Comment 12
"Line 207-209, Is this section part of the template?"
Response 12:
Thank you for your valuable comments. Could you please further explain your comment on template? I would like to understand your point of view better so that I can revise it accordingly.
Comment 13
"Lines 213-214: Chromatography and ultracentrifugation not mentioned in Methods."
Response 13:
We supplemented the Methods section.
Revised Text (Section 2.3, Lines 115-117):
"The filtrate was isolated via differential ultracentrifugation (120,000 g, 2 h), and the resulting precipitate was collected to isolate Exos. This precipitation was resuspended to 1 mL of PBS. Exos were further purified by size exclusion chromatography (Exosuper purification kit , ECHO BIOTECH) , yielding 3.5 mL of fractions containing Exos in each purification tube. "
Comment 14
"Lines 228-235. The results in Figure 2A do not match what is described in the text."
Response 14:
Thanks to your comments, we have modified the image in Figure 1A and added experimental details.
Revised Text (Section 3.2, Lines 246-247):
"and M2-type macrophages were further increased after 48 hours."
Comment 15
"In graph 2B it is not clear or mentioned which macrophages were treated with Exo/Exo-LM?"
Response 15:
We added experimental details.
Revised Text (Section 2.6, Lines 149-150):
"...cell supernatants from RAW264.7 macrophages stimulated with EXO and EXO-LM for 24 hours... "
Comment 16
"Throughout the text, two ways of abbreviating Listeria monocytogenes are used: homogenize and use only one."
Response 16:
We standardized all references to L. monocytogenes (italicized) or LM after first use.
- Response to Comments on the Quality of English Language
The manuscript has been professionally edited to improve readability. - Additional clarifications
Figure 1: We replaced Figures 1A and 1C.
Figure 2: We replaced Figures 1A and 2D.
Western Blot Validation: Raw uncropped blots are provided in Supplementary File S1.
We thank you for the rigorous evaluation. All revisions are marked in the tracked-changes version.

Reviewer 2 Report
Comments and Suggestions for Authors
This study investigates the immunomodulatory role of macrophage-derived exosomes (EXO-LM) during Listeria monocytogenes infection. Transcriptome analysis revealed significant alterations in exosomal mRNAs, lncRNAs, circRNAs, and miRNAs. A key finding highlights the interaction between lncRNA Rpl13a-213 and miR-132-3p. The authors demonstrate that Rpl13a-213 inhibits miR-132-3p, promoting pro-inflammatory cytokine expression and reducing bacterial survival. These results provide novel insights into the ceRNA-mediated regulation of immune responses and the mechanisms facilitating L. monocytogenes survival within host cells.
Recommendations:
General
Use consistent terminology for exosomes (e.g., EXO, EXO-LM, Exos).
Introduction
Line 87-88: The authors state, "The tube was then subjected to ultracentrifugation at 160,000 g for 16 hours." Ultracentrifugation at 160,000 g for 16 hours appears excessive for standard exosome isolation and does not align with the protocol described in Section 2.3. Additionally, the temperature conditions for the ultracentrifugation step are not mentioned and should be clarified. No mention of experiments to validate that serum was exosome-free after ultracentrifugation (e.g. NTA). This should be addressed.
Line 91-96: The culture conditions of L monocytogenes are general. Specify whether pH, oxygen, or shaking conditions.
Line 109: The brand of the Exosuper purification kit is not specified and should be included for transparency and reproducibility.
Line 156-179: The description of transcriptomic analysis lacks the imformation to validate sequencing accuracy, such as sequencing depth and sample replicates.
Results
Lines 224-226: The text states, "on the surface of the extracted Exo stocks (Figure 1E)." However, Figure 1E does not exist. It seems the intended reference is Figure 1C, which displays the Western blot analysis, need cross-verification.
Lines 212-226: There is a clear discrepancy between the figures mentioned in the text and the actual images. The figure references and their alignment with the text should be thoroughly reviewed and corrected.
Figure 1: The figure does not align with the corresponding descriptions in the text. Adjustments are necessary to ensure consistency.
Figure 4:
The legends of the volcano plots and heat maps are unclear and difficult to interpret. Provide clearer labels and adjust for readability.
The reference to Figure 4H in the text is not marked in the figure or its legend. Ensure it is accurately labeled and included in the figure legend.
Lines 325-327: "binding sites were experimentally validated using the dual-luciferase reporter assay" is mentioned. However, it would benefit from more quantitative data (e.g., fold change intenssity in luciferase activity).
Discussion
Contrast the findings with previous studies to highlight the novelty would benefit.
Author Response
Response to Reviewer 1
- Summary
Thank you very much for your constructive comments. Your technical acumen and precise feedback on experimental design and data interpretation have been instrumental in elevating the reproducibility and robustness of our results. We deeply appreciate your rigor.We have carefully revised the manuscript based on your suggestions. All modifications are highlighted in the re-submitted files using track changes. Below are detailed responses to each of your comments. - Point-by-point response to Comments and Suggestions for Authors
Comment 1
"Use consistent terminology for exosomes (e.g., EXO, EXO-LM, Exos)."
Response 1:
Thank you for highlighting this inconsistency. We have standardized the terminology throughout the manuscript:
EXO-LM exclusively refers to exosomes from L. monocytogenes-infected macrophages.
EXO denotes exosomes from uninfected macrophages.
The abbreviation "Exos" has been removed.
Revised Text (Abstract, Lines 14-15 and 2.3, Lines 123-124):
"Exosomes secreted by macrophages during Listeria monocytogenes infection (hereafter EXO-LM)" and "Exosomes from uninfected macrophages (hereafter EXO)were extracted using the same method."
Comment 2
"Line 87-88: The authors state, "The tube was then subjected to ultracentrifugation at 160,000 g for 16 hours." Ultracentrifugation at 160,000 g for 16 hours appears excessive for standard exosome isolation and does not align with the protocol described in Section 2.3. Additionally, the temperature conditions for the ultracentrifugation step are not mentioned and should be clarified. No mention of experiments to validate that serum was exosome-free after ultracentrifugation (e.g. NTA). This should be addressed."
Response 2:
We have added the centrifugation temperature and validated exosome removal:
Revised Text (Section 2.1, Lines 89-91):
Incorporating the methodologies employed in the manufacture of select commercially available exosome-free sera, the preparation of exosome-free serum necessitated an extended period of ultracentrifugation (160,000 g, 16 h) to effectively eliminate pre-existing vesicles. EXO-LM isolation used shorter centrifugation (120,000 g, 2 h) to preserve vesicle integrity.
"The filtrate was isolated via differential ultracentrifugation at 160,000 g for 16 hours at 4°C , and the resulting precipitate was collected to isolate exosomes. "
Comment 3
"Line 91-96: The culture conditions of L. monocytogenes are general. Specify whether pH, oxygen, or shaking conditions."
Response 3:
We have added detailed culture conditions:
Revised Text (Section 2.2, Lines 105-107):
"L. monocytogenes was cultured in BHI broth (pH 7.2 ± 0.2) at 37°C under aerobic conditions with shaking at 200 rpm, and *E. coli* DH5α was grown in LB broth (pH 7.0 ± 0.2) at 37°C under aerobic conditions with shaking at 200 rpm."
Comment 4
"Line 109: The brand of the Exosuper purification kit is not specified and should be included for transparency and reproducibility."
Response 4:
The kit manufacturer has been explicitly stated:
Revised Text (Section 2.3, Lines 119):
"This precipitation was resuspended to 1 mL of PBS. exosomes were further purified by size exclusion chromatography (Exosuper purification kit , ECHO BIOTECH) ,..."
Comment 5
"Line 156-179: The description of transcriptomic analysis lacks the imformation to validate sequencing accuracy, such as sequencing depth and sample replicates."
Response 5:
Sequencing parameters and replicates are now included:
Revised Text (Section 2.8, Lines 177-178):
"These libraries were then sequenced on the Illumina NovaSeq platform, producing 150 bp paired end reads(average depth: 30× per sample). Three independent biological replicates were analyzed for each group."
Comment 6
"Lines 224-226: The text states, "on the surface of the extracted Exo stocks (Figure 1E)." However, Figure 1E does not exist. It seems the intended reference is Figure 1C, which displays the Western blot analysis, need cross-verification."
Response 6:
The erroneous reference has been corrected:
Revised Text (Section 3.1, Lines 244):
"Western blot analysis validated exosomal markers CD9, CD63, and TSG101 (Figure 1C)."
Comment 7
"Lines 212-226: There is a clear discrepancy between the figures mentioned in the text and the actual images. The figure references and their alignment with the text should be thoroughly reviewed and corrected."
Response 7:
We have revised the figure legends for clarity:
Revised Figure 4 Legend:
"Volcano plots show differentially expressed RNAs (red: upregulated; blue: downregulated; dashed lines: fold change ≥2, FDR <0.05). Heatmaps display hierarchical clustering of top 50 genes (color scale: log2 fold change)."
Comment 7
"The reference to Figure 4H in the text is not marked in the figure or its legend. Ensure it is accurately labeled and included in the figure legend."
Response 7:
We've changed the error text:
Revised Text (Section 3.4, Lines 343):
"...that this molecular pair plays a crucial role in macrophage immune function (Figure 4G). "
Comment 8
"Lines 325-327: "binding sites were experimentally validated using the dual-luciferase reporter assay" is mentioned. However, it would benefit from more quantitative data (e.g., fold change intenssity in luciferase activity)."
Response 8:
Quantitative data have been added:
Revised Text (Section 3.5, Lines 330-332):
"Overexpression of Rpl13a-213 reduced luciferase activity by 62.3% ± 5.1% compared to the mutant control (P < 0.01, Figure 5A)."
Comment 9
"Contrast the findings with previous studies to highlight the novelty would benefit."
Response 9:
Given the limited research in this field, relevant literature has been cited in the Introduction section. Additionally, a comparative analysis was incorporated into the Discussion to contextualize the findings within the existing body of literature.
Revised Text (Section 4, Lines 415-418):
" Consistent with previous findings by Nandakumar et al.[18] and Kaletka et al.[20], L. monocytogenes has been shown to manipulate host-derived exosomes as a strategy to subvert immune surveillance, further corroborating the critical role of extracellular vesicle-mediated mechanisms in bacterial immune evasion."
- Response to Comments on the Quality of English Language
The manuscript has been professionally edited to ensure clarity and grammatical accuracy. - Additional clarifications
Figure 1: We replaced Figures 1A and 1C.
Figure 2: We replaced Figures 1A and 2D.
Western Blot Validation: Raw uncropped blots are provided in Supplementary File S1.
We thank Reviewer 1 for the thorough critique, which has significantly improved the manuscript. All revisions are highlighted in the tracked-changes version.

Round 2
Reviewer 1 Report
Comments and Suggestions for Authors
The authors responded to all my observations, I have no further comments.